# In-Context Planning with Latent Temporal Abstractions

## Abstract

Planning-based reinforcement learning for continuous control is bottlenecked by two practical issues: planning at primitive time scales leads to prohibitive branching and long horizons, while real environments are frequently partially observable and exhibit regime shifts that invalidate stationary, fully observed dynamics assumptions. We introduce I-TAP (In-Context Latent Temporal-Abstraction Planner), an offline RL framework that unifies in-context adaptation with online planning in a learned discrete temporal-abstraction space. From offline trajectories, I-TAP learns an observation-conditioned residual-quantization VAE that compresses each observation–macro-action segment into a coarse-to-fine stack of discrete residual tokens, and a temporal Transformer that autoregressively predicts these token stacks from a short recent history. The resulting sequence model acts simultaneously as a context-conditioned prior over abstract actions and a latent dynamics model. At test time, I-TAP performs Monte Carlo Tree Search directly in token space, using short histories for implicit adaptation without gradient updates, and decodes selected token stacks into executable actions. Across deterministic MuJoCo, stochastic MuJoCo with different latent dynamics regimes, and high-dimensional Adroit manipulation, including partially observable variants, I-TAP is competitive with strong model-free and model-based baselines and achieves the best aggregate performance in stochastic and partially observable settings.

## 1 Introduction

Planning-based reinforcement learning (RL) has delivered strong results in discrete decision-making domains (e.g., board games and video games) (Silver et al., 2017; Schrittwieser et al., 2020) and has shown increasing promise in continuous control (Hubert et al., 2021). However, planning-based offline RL for continuous control faces two practical challenges. First, many real environments are effectively partially observable due to latent parameters (e.g., unobserved disturbances or payload changes), which breaks the stationary, full observability assumptions commonly adopted by learned dynamics models used for planning. When these latent factors are not properly handled during planning, they manifest as apparent stochasticity from the planner's perspective (Antonoglou et al., 2022), exacerbating planning complexity. Second, planning over primitive continuous actions induces large branching factors and long effective horizons, making search expensive especially under uncertainty.

To address adaptation under partial observability, recent work has reframed RL as sequence modeling, leveraging Transformers (Vaswani et al., 2017) trained on trajectories to produce policies conditioned on a finite context window (Chen et al., 2021; Brown et al., 2020; Furuta et al., 2022; Liu & Abbeel, 2023; Laskin et al., 2023; Huang et al., 2024). Conditioning on recent interaction history enables in-context adaptation without test-time gradient updates, and can implicitly capture latent task variables when they are identifiable from history. Yet these sequence-model policies are typically deployed as direct action predictors, which introduces two limitations: (i) Without an explicit decision-time optimizer, they often inherit the constraints and suboptimalities of the offline dataset (Son et al., 2025). (ii) In stochastic environments, converting a predictive model into an optimized decision rule is nontrivial when the model is used only as a conditional policy (Paster et al., 2022). These issues motivate combining in-context models with explicit planning.

Nevertheless, planning directly in raw continuous action space can be inefficient and inflexible (Jiang et al., 2023). Recent planning-based methods reduce complexity by learning temporal abstractions (e.g.,

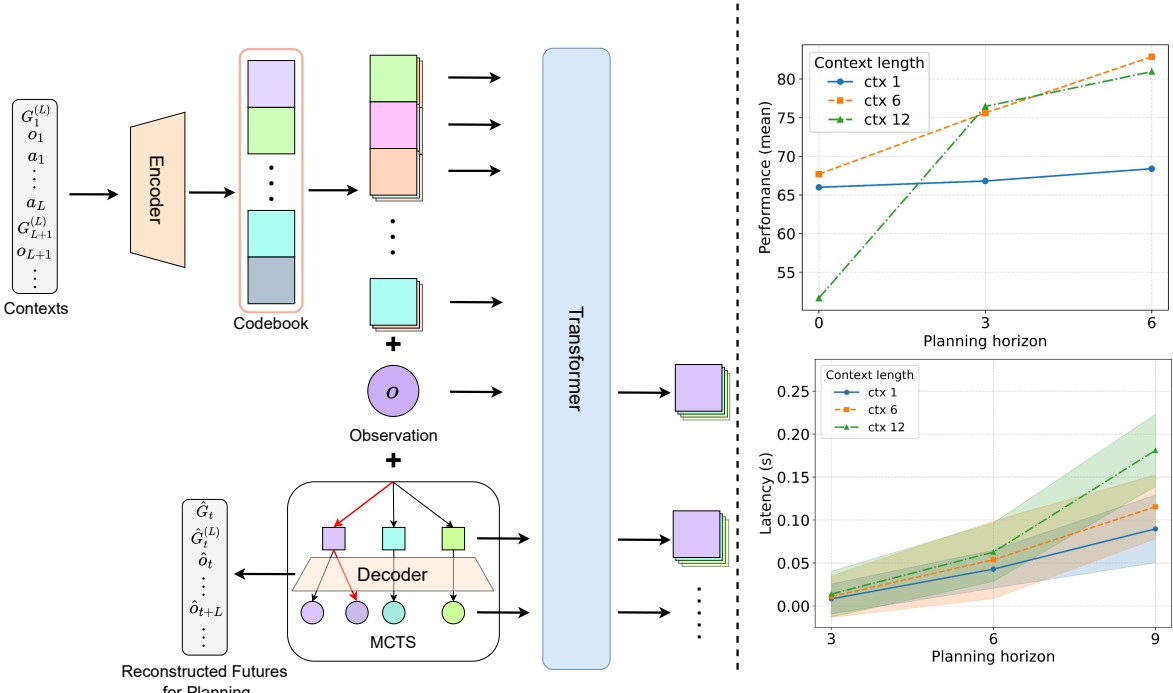

Figure 1: Overview of I-TAP. **Left:** A residual-quantized VAE (RQ-VAE) discretizes continuous observation–action trajectories into a coarse-to-fine token stack. **Right:** Normalized return and per-decision latency as functions of planning horizon and context size on Stochastic MuJoCo, highlighting the importance of a properly sized context window for effective in-context planning under environmental stochasticity and partial observability.

options (Sutton et al., 1999) or macro-actions (Mcgovern & Sutton, 1998)) and planning over high-level decisions (Jiang et al., 2023; Luo et al., 2025), which shortens effective horizons and reduces branching. However, these planners are developed under the assumption of a stationary, fully observed Markov decision process (MDP) and learn dynamics models that condition only on the current state. As a result, state aliasing would create apparent stochasticity and increase the search burden.

Motivated by these challenges, we introduce an in-context planning framework that adapts to different latent parameters in a learned discrete latent temporal abstraction space for continuous control under stochastic dynamics. Our premise is that integrating temporal abstraction with in-context adaptation for planning-based RL addresses these issues jointly. Adapting from recent history and planning conditioned on it in a latent temporal abstraction space allows an agent to: (i) decouple adaptation and planning from the native temporal granularity of MDP, thereby shortening the required context, easing the learning of a reliable sequence model prior by modeling a simpler distribution over discrete latent temporal abstractions instead of high-dimensional continuous actions, and reducing the branching factor during planning; (ii) use context to infer the environment's dynamics (e.g., unobserved perturbation forces), enabling effective adaptation and forecasting across scenario shifts; and (iii) employ an online planner such as Monte Carlo Tree Search (MCTS) for optimization, providing a mechanism to deviate from suboptimal behavior policies in the offline data and handle uncertainty.

To this end, we propose *In-Context Latent Temporal Abstraction Planner* (I-TAP), which learns a discrete latent temporal-abstraction space and a Transformer-based in-context sequence prior over discrete latent codes from offline data, and performs decision-time planning with these models to enable adaptive decision-making. To improve flexibility and scalability in handling high-dimensional continuous observation-action spaces, we adopt an observation-conditioned residual-quantized VAE (RQ-VAE) to learn this discrete latent

temporal-abstraction space, as illustrated in Fig. 1. RQ-VAE (Lee et al., 2022) encodes each macro-step into a depth-$D$ coarse-to-fine stack of code indices drawn from a codebook, providing a compositional discretization with substantially higher effective capacity than a single code. This retains high-fidelity decoding while enabling compact discrete representations, yielding a scalable interface for in-context sequence modeling over abstractions and planning in latent space. MCTS then operates over these latent tokens using context-guided priors to balance exploration and exploitation under uncertainty; finally, we decode the selected latent stack to a primitive action sequence and execute the first action.

Our experiments demonstrate that I-TAP can be trained as a single offline model across behavior policies of varying quality and multiple latent parameters, and evaluated from deterministic to highly stochastic environments. Across these settings, I-TAP matches or outperforms strong offline RL and planning-based baselines, while exhibiting in-context adaptation under partial observability and scaling to high-dimensional continuous control through latent temporal abstractions. In summary, we make the following contributions: (i) We propose I-TAP, an offline RL method that unifies in-context adaptation and online planning for stochastic, partially observable continuous control by planning in a learned latent temporal-abstraction space; (ii) To improve scalability and flexibility, we learn a residual-quantized temporal abstraction and a context-conditioned latent dynamics model from offline trajectories, via an observation-conditioned RQ-VAE and an autoregressive temporal Transformer; (iii) We instantiate an in-context planner that performs Monte Carlo Tree Search directly over latent tokens, reducing the effective branching factor and decision horizon, and enabling improvements beyond suboptimal behavior policies.

## 2 Problem Formulation

**POMDP.** We consider a partially observable Markov decision process (POMDP) $M = (\mathcal{S}, \mathcal{A}, \mathcal{O}, P, R, \gamma)$, where $P(s_{t+1} \mid s_t, a_t)$ is the transition kernel, $R(r_t \mid s_t, a_t)$ is the reward distribution, and $\gamma \in [0, 1)$ is the discount factor. At each time step $t$, the agent receives an observation $o_t \in \mathcal{O}$, selects an action $a_t \in \mathcal{A}$, the environment transitions according to $P(s_{t+1} \mid s_t, a_t)$, and the agent receives a reward $r_t \sim R(\cdot \mid s_t, a_t)$. For a fixed context length $c$, we define the agent's context prior to observing $o_t$ as $c_t = (o_{t-c}, a_{t-c}, r_{t-c}, \ldots, o_{t-1}, a_{t-1}, r_{t-1}) \in \mathcal{C}$. The goal is to maximize the expected discounted return $J = \mathbb{E}_{\pi, M}\left[\sum_{t=0}^{T-1} \gamma^t r_t\right]$ over horizon $T$.

**Meta-RL.** We define a context-conditioned policy $\pi : \mathcal{C} \times \mathcal{O} \to \Delta(\mathcal{A})$, where $\mathcal{C}$ is a fixed-length context space and $\Delta(\mathcal{A})$ denotes distributions over actions. At time $t$, the algorithm forms a bounded context $c_t \in \mathcal{C}$ from the interaction history $h_t$ (e.g., a sliding window of recent transitions) and samples actions as $a_t \sim f(c_t, o_t)$. Meta-RL aims to learn an algorithm $\pi$ that maximizes $J$ in expectation over a distribution of tasks (POMDPs) $p(M)$: $\max_{\pi} \mathbb{E}_{M \sim p(M)}\left[\mathbb{E}_{\pi, M}\left[\sum_{t=0}^{H-1} \gamma^t r_t\right]\right]$. In our setting, each $M$ is induced by an unobserved latent parameter. In offline meta-RL, we assume access to a dataset of trajectories collected by some behavior algorithms on meta-training tasks, from which the context $c_t$ is constructed.

## 3 Background

**Residual-Quantized Variational Autoencoder (RQ-VAE).** Residual-Quantized VAE (RQ-VAE) (Lee et al., 2022) is a discrete autoencoding model that replaces the single-step vector quantization in VQ-based models with residual quantization, representing each latent as a stack of $D$ discrete codes. Let $z_t \in \mathbb{R}^d$ denote a continuous feature vector produced by an encoder, and let the codebook be $E = \{e_1, \ldots, e_K\} \subset \mathbb{R}^d$. RQ-VAE iteratively quantizes residuals in a coarse-to-fine manner by initializing $r_t^{(0)} := z_t$ and, for depths $\ell = 1, \ldots, D$, computing

$$k_{t,\ell} = \arg\min_{k \in [K]} \|r_t^{(\ell-1)} - e_k\|_2^2, \qquad r_t^{(\ell)} = r_t^{(\ell-1)} - e_{k_{t,\ell}},$$

which yields the depth-$\ell$ partial sum $\hat{z}_t^{(\ell)} := \sum_{j=1}^{\ell} e_{k_{t,j}}$ and the final quantized vector $\hat{z}_t := \hat{z}_t^{(D)}$. Using a shared codebook across depths yields an effective representational capacity that grows as $K^D$ without

enlarging $K$. The decoder reconstructs the input from $\hat{z}_t$, and the model is trained jointly with minimizing a reconstruction loss plus a depth-wise commitment regularizer, using the straight-through estimator for backpropagation through discrete assignments and updating the codebook via exponential moving average (EMA) (van den Oord et al., 2017). An RQ-Transformer (Lee et al., 2022) combines a spatial Transformer that summarizes codes from previous positions into a context vector with a depth Transformer that, conditioned on this context, predicts the $D$ codes within the current stack in a coarse-to-fine manner, thereby modeling an autoregressive prior over code stacks.

**Temporal Abstraction via Macro-Actions.** We use the standard macro-action abstraction with a fixed macro length $L$ and a macro-action space $\mathcal{A}^{(L)}$, where each macro-action $m_b \in \mathcal{A}^{(L)}$ consists of $L$ primitive actions. Decisions are made at macro-steps $b = 0, 1, \ldots, N - 1$, corresponding to primitive times $t = bL$, so the episode horizon satisfies $T = NL$. At macro-step $b$, the agent selects $m_b$ and executes it for $L$ primitive steps, generating rewards $\{r_{bL}, \ldots, r_{bL+L-1}\}$.

**Trajectory Representation.** Consider an episode with $T = NL$ primitive steps and macro boundaries at $t = bL$. The return-to-go from primitive time $t$ is $G_t = \sum_{i=t}^{T-1} \gamma^{i-t} r_i$, and the $L$-step discounted return from a macro boundary is $G_{bL}^{(L)} = \sum_{i=0}^{L-1} \gamma^i r_{bL+i}$. We write the macro-level trajectory as $\tau = \left( (G_{bL}, G_{bL}^{(L)}, o_{bL}, m_b) \right)_{b=0}^{N-1}$, where $o_{bL}$ is the observation at the macro boundary and $m_b$ is the macro-action executed for the subsequent $L$ primitive steps. This representation preserves the underlying primitive dynamics while exposing temporally extended actions.

# 4 Method

I-TAP can be viewed as a four-stage pipeline. First, instead of planning over individual continuous actions $a_t$, we group $L$ consecutive primitive actions into a macro-action $m_b = (a_{bL}, \ldots, a_{bL+L-1})$. This reduces the effective planning horizon from primitive time steps to macro time steps. Second, because $m_b$ is still continuous and high-dimensional, we encode each observation–macro-action segment into a discrete latent representation. In our case, this representation is a residual-code stack, where multiple discrete codes jointly describe one macro-step. Third, we train a Transformer to predict these residual-code stacks from the recent history, so the model acts as both a context-conditioned prior over likely macro-actions and a latent dynamics model for possible future outcomes. Finally, at test time, MCTS searches over candidate residual-code stacks rather than raw continuous actions. The selected code stack is decoded into a primitive action sequence, and the agent executes the corresponding action while replanning at the next decision point.

## 4.1 In-Context Residual Discretization of Observation Macro Action Sequences

In-context RL approaches tend to replicate the suboptimal behaviors of the source algorithm (Son et al., 2025), necessitating the integration of a planning mechanism to deviate from suboptimal decisions. Meanwhile, a discrete action space simplifies the representation of action distributions and facilitates the use of advanced planning algorithms (Silver et al., 2018).

To leverage these advantages, prior work has proposed state-conditioned Vector Quantized VAEs (VQ-VAEs) (Jiang et al., 2023) to discretize continuous control into a compact latent action space. However, single-code vector quantization can become a bottleneck: to maintain reconstruction fidelity for high-dimensional features, one often needs either a very large codebook (inflating the discrete vocabulary) or additional compression, both of which can degrade reconstruction quality when features are high-dimensional (Lee et al., 2022; Jiang et al., 2024). In our setting, each macro-step token must summarize an observation together with a temporally extended action, and the encoder processes these as context-dependent sequences, making the embeddings more context-dependent and diverse, which further increases the required quantization capacity. We therefore adopt an observation-conditioned RQ-VAE, which represents each macro-step with a depth-$D$ coarse-to-fine stack of codes, providing substantially higher effective capacity without requiring an excessively large single-step codebook. This yields compact discrete representations while preserving decoding fidelity, which is crucial when downstream decision making depends on model predictions over these abstractions.

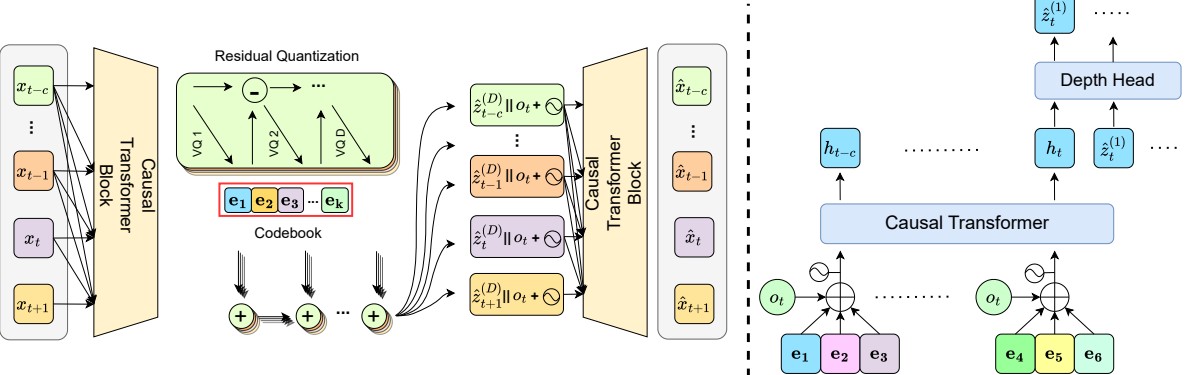

Figure 2: An overview of our RQ-VAE model that discretizes state-macro action sequences and temporal prior for I-TAP

**Tokens and masking.** At macro index $b$, a token is $x_{t_b} = \left(G_{t_b}, G_{t_b}^{(L)}, o_{t_b}, m_{t_b}\right)$. During training, we sample contiguous chunks $(x_{t_{b-c}}, x_{t_{b-c+1}}, \ldots, x_{t_b}, x_{t_{b+1}})$, where the first $c$ tokens provide context and the last two tokens correspond to the current and next macro-step.

Inspired by Luo et al. (2025), we mask return components to reduce variance induced by behavior-policy quality and environmental stochasticity. Specifically, we mask the return-to-go $G_t$ at all positions, since it is a high-variance long-horizon signal and requires choosing a test-time target value. We retain the short-horizon return $G_t^{(L)}$ only in the context tokens and mask it for the current and next tokens $(x_{t_b}, x_{t_{b+1}})$. Consequently, the code assigned to the current macro decision is determined by the observation–macro-action pair and the preceding context, rather than by the realized return, which reduces sensitivity to return variability in the offline data.

This design avoids manually choosing a target $G_{t_b}$, which is susceptible to luck-induced variability in stochastic environments (Paster et al., 2022), while still using $G^{(L)}$ in the context as a stable short-horizon signal for inferring latent regimes.

**Discretization with RQ-VAE** A causal Transformer maps the chunk to per-token features $Z = \left(z_{t_{b-c}}, \ldots, z_{t_b}, z_{t_{b+1}}\right)$. For each token index $t \in \{t_b - c, \ldots, t_b + 1\}$, we apply residual quantization (section 3) with a shared codebook $E = \{e_1, \ldots, e_K\} \subset \mathbb{R}^d$ to obtain a depth-$D$ code stack $k_{t,1:D}$ and partial sums $\hat{z}_t^{(\ell)} = \sum_{j=1}^{\ell} e_{k_{t,j}}$; we use the final quantized embedding $\hat{z}_t^{(D)}$ as the discrete representation of $x_t$.

Motivated by evidence from prior work (Jiang et al., 2023; Luo et al., 2023) that state-conditioned decoding allows more compact codebooks while preserving reconstruction fidelity. For each token, our decoder conditions on $o_{t_b}$ and the quantized latent $\hat{z}_t^{(D)}$ via a linear adapter and causal attention, reconstructing all features for every token in the chunk by leveraging its previous context: $(\hat{G}_{t_{b-c}}, \hat{G}_{t_{b-c}}^{(L)}, \hat{o}_{t_{b-c}}, \hat{m}_{t_{b-c}}, \ldots, \hat{G}_{t_{b+1}}, \hat{G}_{t_{b+1}}^{(L)}, \hat{o}_{t_{b+1}}, \hat{m}_{t_{b+1}})$. We optimize a reconstruction loss plus a residual partial-sum commitment:

$$\mathcal{L} = \sum_{\tau} \alpha_\tau \left\| (\hat{x}_\tau - x_\tau) \right\|_2^2 + \frac{\beta_{\text{ps}}}{D} \sum_{\ell=1}^{D} \left\| Z - \text{sg}\left[ \hat{Z}^{(\ell)} \right] \right\|_2^2.$$

Here $\alpha_\tau = \alpha_{\text{tail}}$ for the last two tokens $(x_{t_b}, x_{t_{b+1}})$ and $\alpha_{\text{ctx}}$ otherwise; $\hat{Z}^{(\ell)}$ is the depth-wise $\ell$ partial sum of residual code embeddings; and $\text{sg}[\cdot]$ denotes stop-gradient. The depth-wise partial-sum term stabilizes residual quantization and prevents code hopping across depths, consistent with RQ-VAE (Lee et al., 2022).

## 4.2 Temporal Prior over Residual Code Stacks

Let $k_{t,1:D}$ denote the depth-$1:D$ codes at macro time $t$. We learn an observation conditioned depth-aware autoregressive prior that factorizes across time and within-time depth:

$$p_\phi\big(k_{t,1:D}\,\big|\,k_{<t,1:D},\,o_t\big) \;=\; \prod_{\ell=1}^{D} p_\phi\big(k_{t,\ell}\,\big|\,k_{<t,1:D},\,k_{t,<\ell},\,o_t\big).$$

Inspired by the factorized residual-code prior in RQ-Transformer (Lee et al., 2022), we design an efficient prior for planning that avoids expanding a depth-$D$ code stack into $D$ separate tokens per time step (which would increase context length from $C$ to $CD$). Specifically, a temporal trunk (causal over $t$) embeds each past position by the sum of its depth embeddings $\sum_{j=1}^{D} e_{k_{u,j}}$, adds positional and observation embeddings, and produces a context $h_t$. We use a lightweight depth head (a shared Multi-Layer Perceptron) to predict $k_{t,1}, k_{t,2}, \ldots, k_{t,D}$ by conditioning on $h_t$, a depth embedding, and the partial sum of shallower depths at time $t$. We then minimize the negative log-likelihood loss:

$$\mathcal{L}_{\text{prior}} \;=\; \mathbb{E}\Big[-\sum_{t}\sum_{\ell=1}^{D} \log p_\phi\big(k_{t,\ell}\,\big|\,k_{<t,1:D},\,k_{t,<\ell},\,o_t\big)\Big].$$

The time and depth factorization retains long temporal context while modeling within-position coarse to fine refinement efficiently.

## 4.3 In-Context Planning With Monte Carlo Tree Search

Prior work has leveraged MCTS to mitigate stochasticity arising from the environment in both online (Antonoglou et al., 2022) and offline RL (Luo et al., 2025). Planning in the real world, however, poses two additional challenges. First, partial observability induces apparent stochasticity when the context cannot reliably disambiguate latent states. Second, one needs a mechanism to balance the inherited bias and exploration at decision time when policies are learned from suboptimal behavior.

We therefore adopt MCTS as the online planner in our latent temporal-abstraction space (Fig. 3). Using the learned latent dynamics to sample multiple future continuations and backing up their returns, MCTS estimates the expected value of each candidate latent action, making action selection less sensitive to noisy single-rollout return estimates. Moreover, by planning over a context-conditioned latent state, the search mitigates apparent stochasticity from partial observability (to the extent captured by the context). Finally, coupling P-UCT (Silver et al., 2017) with our context-conditioned prior over latent tokens enables targeted exploration and provides a principled way to override suboptimal priors when predicted returns justify it, while keeping the search within the distribution supported by the learned model.

**Latent Search Tree.** At time $t$, the agent observes $o_t$ and an interaction history in a sliding window of length $L{\times}c$. We encode this history into residual-quantized codes to obtain a context window $k_{t-1:t-c,\,1:D}$. A *decision node* is $s = (o_t, k_{t-1:t-c,1:D})$, and an *action edge* $a$ out of $s$ is a depth $D$ code stack $k_{t,1:D}$. Executing $a$ produces a distribution over *outcome codes* $k_{t+1,1:D}$ via our temporal prior $p_\phi(k_{t+1,1:D} \mid k_{t,1:D}, o_t, k_{t-1:t-c,1:D})$, and each outcome is decoded to a tail $(\hat{G}_{t+L}, \hat{G}_{t+L}^{(L)}, \hat{o}_{t+L}, \hat{m}_{t+L})$, yielding the successor decision node $s_{t+L}$. To mitigate the cost of iterative model calls with context (a bottleneck when parallelism is underused), following Luo et al. (2025) we pre-construct a context-conditioned latent search tree by parallel sampling and caching a finite set of high-probability action stacks and their predicted outcomes from $p_\phi$, and run MCTS over this cached search space (section A.1).

**Policy-guided selection** From node $s$, we restrict top-$K$ action candidates and use a behavior-like prior to prioritize searching in-distribution actions without sacrificing exploration. Let the policy head produce probability logits $l_a$, we select $a$ by the AlphaZero-style P-UCT score:

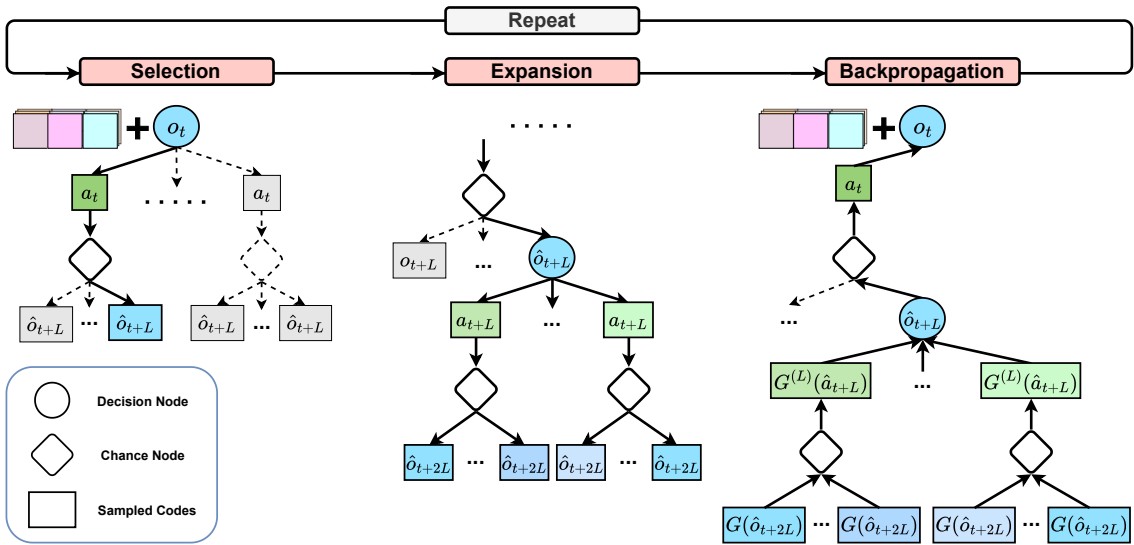

Figure 3: Macro-level MCTS overview. Each iteration uses P-UCT to select a macro-action, expands several candidates and their predicted outcomes in parallel, then backs up the resulting Q-estimates through the search tree to steer subsequent exploration.

$$a = \arg\max_a \Big[ Q(s,a) \; + \; \underbrace{\Big(c_1 + \log\frac{N(s) + c_2 + 1}{c_2}\Big) \frac{\sqrt{N(s)}}{1 + N(s,a)}}_{\text{exploration term}} \underbrace{\Big(\pi_T(a \mid s) = \frac{e^{l_a/T}}{\sum_{b \in \mathcal{A}(s)} e^{l_b/T}}\Big)}_{\text{temperature scaled prior}} \Big],$$

where $N(s)$ and $N(s,a)$ are visit counts and $Q(s,a)$ is the action value; $c_1, c_2 > 0$ are P-UCT exploration constants. This coupling of a learned prior with P-UCT emphasizes in-distribution actions (large $\pi_T$) without sacrificing exploration via the exploration term, allowing the search to deviate from the source policy when returns warrant it.

**Parallel expansion and backpropagation.** At each decision node $s_t$ we expand the top-$K$ candidates in parallel, sample $D$ outcome codes at the chance node, and decode them to obtain a successor $s_{t+L} = (\hat{o}_{t+L}, k'_{t:t-c+1, 1:D})$ and its leaf value. We then back up $Q(s,a)$ along the visited path using incremental averages and update visit counts.

## 5 Experiments

We evaluate I-TAP through comprehensive empirical studies using tasks from the D4RL benchmark (Fu et al., 2020), focusing on standard Gym locomotion tasks and complex high-dimensional Adroit robotic manipulation. Our experiments assess both performance and adaptability of I-TAP across varying degrees of environmental stochasticity. We additionally conduct ablation studies to examine how macro-action length, context length, planning horizon, and residual depth affect performance. We further analyze the relationship between decision latency and context length in Section 5.3.

**Baselines.** We compare I-TAP to strong offline RL baselines: model-free actor–critic methods Conservative Q-Learning (CQL; Kumar et al. (2020)), Implicit Q-Learning (IQL; Kostrikov et al. (2022)), and Flow Q-Learning (FQL; Park et al. (2025)); context-conditioned policy methods such as Decision Transformer (DT; Chen et al. (2021)) and Long-Short Decision Transformer (LSDT; Wang et al. (2025)); and model-based planners that operate over learned temporal abstractions, including Trajectory Autoencoding Planner (TAP; Jiang et al. (2023)), Latent Macro-Action Planner (L-MAP; Luo et al. (2025)), where TAP is largely insensitive to raw action dimensionality and shows strong performance on high-dimensional Adroit manipulation, and

L-MAP likewise scales well to high-dimensional control while remaining robust under stochastic dynamics. Finally, a risk-sensitive, model-based specialist for stochastic domains (1R2R; Rigter et al. (2023)).

**Experimental Setup.** To assess I-TAP's adaptation capabilities across varying latent task parameters and associated dynamics, we conduct comprehensive experiments using the Stochastic MuJoCo tasks introduced by Rigter et al. (2023). Each environment defines an unobserved global latent task parameter controlling perturbation levels (0, 2.5, 5), which in turn specify the distribution of instantaneous hidden forces at each step. For each behavior-policy dataset type, we train a single model per method (I-TAP, DT, L-MAP, LSDT, and FQL) on the same pooled data across perturbation regimes. At test time, these methods perform no gradient updates; I-TAP, DT, and LSDT adapt through conditioning on the context window, while L-MAP and FQL are evaluated under the same pooled-training protocol but do not perform context-conditioned test-time planning. For the remaining standard offline RL baselines (e.g., CQL/IQL), we use the corresponding stochastic-domain results from prior work under the same benchmark protocol. To test scalability to high-dimensional control and partial observability, we evaluate on Adroit (Rajeswaran et al., 2018) in (i) the original fully observable setting and (ii) a partially observable variant where we mask a subset of target-position coordinates (see Appendix A.2 for the exact dimensions and masking procedure). Unless otherwise noted, we set the latent context size to $C = 6$ tokens; with a macro length of $L = 3$, this window summarizes 18 past primitive transitions. Further domain and hyperparameter details appear in Appendix A.2.

## 5.1 Main results

Table 1: Normalised results for high-noise (5), moderate-noise (2.5), deterministic (0) environments. Bold numbers indicate the best reported mean in each row and are not used as a statistical-significance marker.

| Dataset Type | Env (Noise) | Model-Based | | | | Model-Free | | | | |
|---|---|---|---|---|---|---|---|---|---|---|
| | | I-TAP | L-MAP | TAP | 1R2R | DT | LSDT | FQL | CQL | IQL |
| Medium-Expert | Hopper (5) | **83.11 ± 3.13** | 71.49 ± 3.46 | 37.31 ± 3.66 | 37.99 ± 2.71 | 61.87 ± 2.56 | 66.97 ± 8.56 | 49.76 ± 2.16 | 68.03 ± 3.94 | 44.83 ± 2.58 |
| | Hopper (2.5) | 103.59 ± 2.37 | 93.40 ± 3.65 | 40.86 ± 5.42 | 52.19 ± 8.37 | 73.86 ± 2.68 | 78.21 ± 17.24 | 47.52 ± 5.81 | **106.17 ± 2.16** | 60.61 ± 3.46 |
| | Hopper (0) | **111.40 ± 0.48** | 106.74 ± 2.24 | 85.55 ± 3.83 | 57.40 ± 6.06 | 101.6 ± 1.85 | 81.81 ± 18.84 | 47.21 ± 4.52 | 105.4 | 91.5 |
| | Walker2D (12) | 92.31 ± 2.22 | **92.75 ± 1.84** | 91.09 ± 2.78 | 32.38 ± 4.55 | 52.42 ± 1.27 | 39.62 ± 2.69 | 66.78 ± 6.21 | 83.18 ± 3.70 | 68.61 ± 3.33 |
| | Walker2D (7) | **97.55 ± 1.46** | 93.48 ± 1.20 | 91.40 ± 1.42 | 56.48 ± 7.51 | 64.67 ± 1.00 | 52.17 ± 3.11 | 75.26 ± 6.10 | 91.44 ± 1.44 | 86.66 ± 1.84 |
| | Walker2D (0) | 100.21 ± 0.64 | 100.38 ± 0.72 | 105.32 ± 2.03 | 73.18 ± 6.29 | 64.65 ± 0.79 | 58.40 ± 5.85 | 74.23 ± 3.62 | 108.8 | **109.6** |
| Mean (Medium-Expert) | | **98.03** | 93.04 | 75.26 | 51.60 | 69.85 | 62.86 | 60.13 | 93.84 | 76.97 |
| Medium | Hopper (5) | **67.20 ± 2.10** | 59.05 ± 2.93 | 43.93 ± 2.66 | 33.99 ± 0.92 | 55.91 ± 2.02 | 54.84 ± 3.41 | 39.23 ± 2.21 | 45.21 ± 2.97 | 49.69 ± 2.47 |
| | Hopper (2.5) | **71.20 ± 2.30** | 63.21 ± 3.10 | 43.64 ± 2.25 | 65.24 ± 3.31 | 60.97 ± 0.82 | 55.41 ± 8.87 | 36.40 ± 1.37 | 49.92 ± 3.00 | 56.00 ± 3.60 |
| | Hopper (0) | **81.10 ± 2.20** | 61.65 ± 2.81 | 69.14 ± 2.33 | 55.49 ± 3.99 | 58.14 ± 0.24 | 48.37 ± 9.63 | 34.20 ± 1.91 | 58.0 | 66.3 |
| | Walker2D (12) | 59.50 ± 2.00 | 59.05 ± 2.30 | 52.20 ± 2.76 | 32.13 ± 4.51 | 32.20 ± 0.83 | 38.67 ± 5.09 | 48.47 ± 3.22 | **61.49 ± 3.24** | 47.53 ± 3.05 |
| | Walker2D (7) | **66.00 ± 2.00** | 62.23 ± 1.34 | 44.46 ± 1.82 | 65.16 ± 2.84 | 43.77 ± 0.95 | 46.67 ± 3.89 | 52.96 ± 3.82 | 49.38 ± 2.02 | 48.82 ± 2.31 |
| | Walker2D (0) | **81.40 ± 0.60** | 75.54 ± 1.59 | 51.75 ± 3.30 | 55.69 ± 4.97 | 55.36 ± 0.61 | 56.37 ± 12.36 | 58.04 ± 2.97 | 72.5 | 78.3 |
| Mean (Medium) | | **71.07** | 63.46 | 50.85 | 51.28 | 51.06 | 50.05 | 44.88 | 56.08 | 57.77 |
| Medium-Replay | Hopper (5) | **69.67 ± 2.25** | 60.76 ± 2.79 | 48.69 ± 2.97 | 68.25 ± 3.78 | 35.17 ± 0.96 | 17.24 ± 2.57 | 22.91 ± 3.73 | 51.70 ± 3.09 | 43.27 ± 2.78 |
| | Hopper (2.5) | **83.63 ± 1.77** | 73.81 ± 2.67 | 38.10 ± 3.22 | 22.82 ± 2.08 | 35.76 ± 1.01 | 18.34 ± 3.29 | 25.40 ± 4.20 | 40.53 ± 1.52 | 49.12 ± 3.38 |
| | Hopper (0) | 90.02 ± 1.04 | 90.8 ± 0.63 | 80.92 ± 3.79 | 89.67 ± 1.92 | 43.01 ± 1.36 | 16.45 ± 4.41 | 24.07 ± 4.33 | **95.0** | 94.7 |
| | Walker2D (12) | **69.47 ± 2.07** | 59.16 ± 2.92 | 55.15 ± 3.29 | 65.63 ± 3.41 | 37.22 ± 0.78 | 20.82 ± 3.23 | 31.49 ± 3.48 | 50.33 ± 3.88 | 45.13 ± 2.38 |
| | Walker2D (7) | **74.26 ± 1.69** | 69.20 ± 2.55 | 43.49 ± 2.27 | 52.23 ± 2.22 | 49.51 ± 0.81 | 29.87 ± 4.77 | 37.06 ± 5.19 | 40.24 ± 1.67 | 40.77 ± 2.72 |
| | Walker2D (0) | 77.38 ± 0.97 | 70.66 ± 1.78 | 72.32 ± 3.26 | **90.67 ± 1.98** | 48.44 ± 0.76 | 50.44 ± 6.35 | 47.36 ± 6.74 | 77.2 | 77.2 |
| Mean (Medium-Replay) | | **77.41** | 70.73 | 56.45 | 64.88 | 41.52 | 25.53 | 31.38 | 59.17 | 58.37 |

**Mujoco** We use the Stochastic MuJoCo suite to evaluate I-TAP's in-context adaptation and robustness to uncertainty. Each episode is governed by a latent task parameter that selects the perturbation regime (e.g., 0, 2.5, 5) and thereby determines the distribution of hidden forces. We organize the evaluation into three complementary axes: (i) in-distribution latent-regime adaptation, (ii) held-out latent-regime generalization, and (iii) partial-observability evaluation across diverse Adroit manipulation tasks. For the first two axes, Table 1 evaluates *in-distribution* regimes where latent-parameter values are present in the meta-training datasets, so performance primarily reflects a method's ability to infer the active regime from history, adapt its decisions online, and handle uncertainty. Table 2 evaluates held-out regimes not included in training to evaluate interpolation and extrapolation across latent task parameters.

Across all dataset types, I-TAP achieves the highest mean score and remains strong across changing dynamics without any gradient updates at test time. These gains are consistent with I-TAP's ability to (i) leverage trajectory history for in-context identification of the latent regime and (ii) plan online to optimize actions instead of merely imitating the behavior policy in the dataset. We interpret small row-level gaps conservatively,

Table 2: Normalised results (mean ± std). Noise parameter values are shown in parentheses. Bold indicates the best reported mean in each row and is not used as a statistical-significance marker.

| Dataset Type | Env (Noise) | I-TAP | L-MAP | DT | LSDT | FQL |
|---|---|---|---|---|---|---|
| Medium-Expert | Hopper (7.5) | **69.56 ± 2.72** | 62.90 ± 3.50 | 58.96 ± 0.21 | 67.15 ± 5.30 | 44.62 ± 2.40 |
| | Hopper (3.75) | **90.40 ± 2.98** | 80.51 ± 3.86 | 64.97 ± 0.25 | 75.09 ± 11.84 | 47.18 ± 3.49 |
| | Hopper (1.25) | **110.52 ± 0.80** | 94.65 ± 3.03 | 87.41 ± 0.24 | 83.82 ± 19.03 | 45.82 ± 4.16 |
| | Walker2D (15) | **88.08 ± 2.66** | 87.21 ± 3.42 | 44.07 ± 0.26 | 29.86 ± 4.33 | 58.66 ± 4.00 |
| | Walker2D (9.5) | **95.47 ± 1.90** | 92.31 ± 1.81 | 56.23 ± 0.22 | 47.60 ± 3.98 | 72.16 ± 2.76 |
| | Walker2D (3.5) | **98.29 ± 0.92** | 96.53 ± 0.62 | 65.07 ± 0.18 | 56.70 ± 5.36 | 76.06 ± 4.39 |
| Mean (Medium-Expert) | | **92.05** | 85.69 | 62.79 | 60.04 | 57.42 |
| Medium | Hopper (7.5) | **59.70 ± 2.20** | 55.26 ± 2.53 | 51.62 ± 0.16 | 52.64 ± 4.29 | 39.30 ± 1.74 |
| | Hopper (3.75) | **71.00 ± 2.30** | 66.68 ± 2.50 | 56.09 ± 0.17 | 55.02 ± 5.44 | 37.10 ± 2.07 |
| | Hopper (1.25) | **73.00 ± 2.30** | 71.08 ± 2.56 | 59.39 ± 0.12 | 52.97 ± 9.67 | 33.78 ± 2.36 |
| | Walker2D (15) | **52.30 ± 2.10** | 50.10 ± 2.82 | 30.69 ± 0.18 | 33.89 ± 2.81 | 43.21 ± 4.46 |
| | Walker2D (9.5) | **60.00 ± 2.00** | 55.96 ± 2.53 | 39.16 ± 0.19 | 45.17 ± 3.86 | 53.67 ± 3.12 |
| | Walker2D (3.5) | 74.80 ± 1.30 | **75.77 ± 1.59** | 51.89 ± 0.17 | 53.19 ± 6.15 | 57.21 ± 2.70 |
| Mean (Medium) | | **65.13** | 62.47 | 48.14 | 48.81 | 44.04 |
| Medium-Replay | Hopper (7.5) | **61.21 ± 2.12** | 50.79 ± 2.56 | 30.17 ± 0.14 | 17.00 ± 2.97 | 22.25 ± 2.96 |
| | Hopper (3.75) | **80.17 ± 1.73** | 70.06 ± 2.87 | 33.03 ± 0.18 | 18.22 ± 2.47 | 22.21 ± 2.70 |
| | Hopper (1.25) | **89.16 ± 1.34** | 83.99 ± 2.21 | 39.22 ± 0.24 | 17.61 ± 4.20 | 24.61 ± 3.17 |
| | Walker2D (15) | **58.00 ± 2.07** | 50.64 ± 2.87 | 29.91 ± 0.15 | 20.16 ± 1.87 | 27.10 ± 3.31 |
| | Walker2D (9.5) | **69.75 ± 1.81** | 64.68 ± 2.71 | 42.78 ± 0.16 | 26.20 ± 4.44 | 35.40 ± 4.07 |
| | Walker2D (3.5) | **77.20 ± 1.03** | 71.98 ± 2.12 | 58.49 ± 0.15 | 41.43 ± 7.84 | 43.13 ± 6.40 |
| Mean (Medium-Replay) | | **72.58** | 65.36 | 38.93 | 23.44 | 29.11 |

especially when uncertainty intervals overlap, and focus our discussion on aggregate means and consistent trends across dataset types. Relative to L-MAP, I-TAP's aggregate improvements highlight the benefit of *context-guided* action selection within search: the sequence-model prior steers exploration while the planner evaluates alternatives, rather than relying on MCTS alone to handle uncertainty. DT's absolute performance remains substantially below I-TAP under stochastic dynamics and lower-quality behavior data, highlighting the benefit of combining context-conditioned adaptation with decision-time planning. This is consistent with DT's lack of an integrated planner and its reliance on return conditioning, which can be sensitive to luck-induced variability (Paster et al., 2022). By combining context-based adaptation with downstream planning, I-TAP is less likely to inherit suboptimal dataset behavior and can deviate whenever the planner identifies more promising actions.

Table 3: Adroit robotic hand control results. I-TAP results are reported with five random seeds. Bold indicates the best reported mean in each row and is not used as a statistical-significance marker.

| Dataset Type | Env | Model-Based | | | Model-Free | |
|---|---|---|---|---|---|---|
| | | I-TAP | L-MAP | TAP | LSDT | FQL |
| Cloned | Pen | **88.23 ± 7.42** | 60.68 ± 7.88 | 46.44 ± 7.54 | 23.05 ± 1.24 | 74.00 ± 11.00 |
| Cloned | Hammer | 4.79 ± 1.07 | 2.43 ± 0.29 | 1.32 ± 0.12 | 1.42 ± 0.80 | **11.00 ± 9.00** |
| Cloned | Door | **13.76 ± 1.18** | 13.22 ± 1.34 | 13.45 ± 1.43 | 0.13 ± 0.23 | 2.00 ± 1.00 |
| Cloned | Relocate | 0.10 ± 0.04 | **0.15 ± 0.13** | -0.23 ± 0.01 | -0.23 ± 0.02 | -0.00 ± 0.00 |
| Expert | Pen | 133.62 ± 4.70 | 126.60 ± 5.60 | 127.40 ± 7.70 | 105.38 ± 1.56 | **142.00 ± 6.00** |
| Expert | Hammer | **128.14 ± 0.28** | 127.16 ± 0.29 | 127.60 ± 1.70 | 126.36 ± 0.17 | 125.00 ± 3.00 |
| Expert | Door | **106.08 ± 0.13** | 105.24 ± 0.10 | 104.80 ± 0.80 | 104.20 ± 0.97 | 104.00 ± 1.00 |
| Expert | Relocate | **110.43 ± 0.71** | 107.57 ± 0.76 | 106.21 ± 1.61 | 106.67 ± 0.64 | 107.00 ± 1.00 |
| Expert (POMDP) | Pen | **85.97 ± 7.68** | 69.84 ± 9.81 | 60.87 ± 9.55 | 17.84 ± 4.67 | 34.05 ± 3.09 |
| Expert (POMDP) | Hammer | **69.91 ± 6.87** | 59.21 ± 6.52 | 42.22 ± 12.92 | 69.74 ± 10.46 | 64.46 ± 9.10 |
| Expert (POMDP) | Door | 96.42 ± 2.62 | 89.35 ± 3.41 | 83.71 ± 4.22 | **97.20 ± 2.01** | 83.64 ± 3.78 |
| Expert (POMDP) | Relocate | **50.59 ± 3.91** | 37.36 ± 3.84 | 33.94 ± 3.50 | 0.16 ± 0.10 | 0.33 ± 0.24 |
| **Mean (Expert)** | | **119.57** | 116.64 | 116.50 | 110.65 | 119.50 |
| **Mean (Expert POMDP)** | | **75.72** | 63.94 | 55.19 | 46.24 | 45.62 |

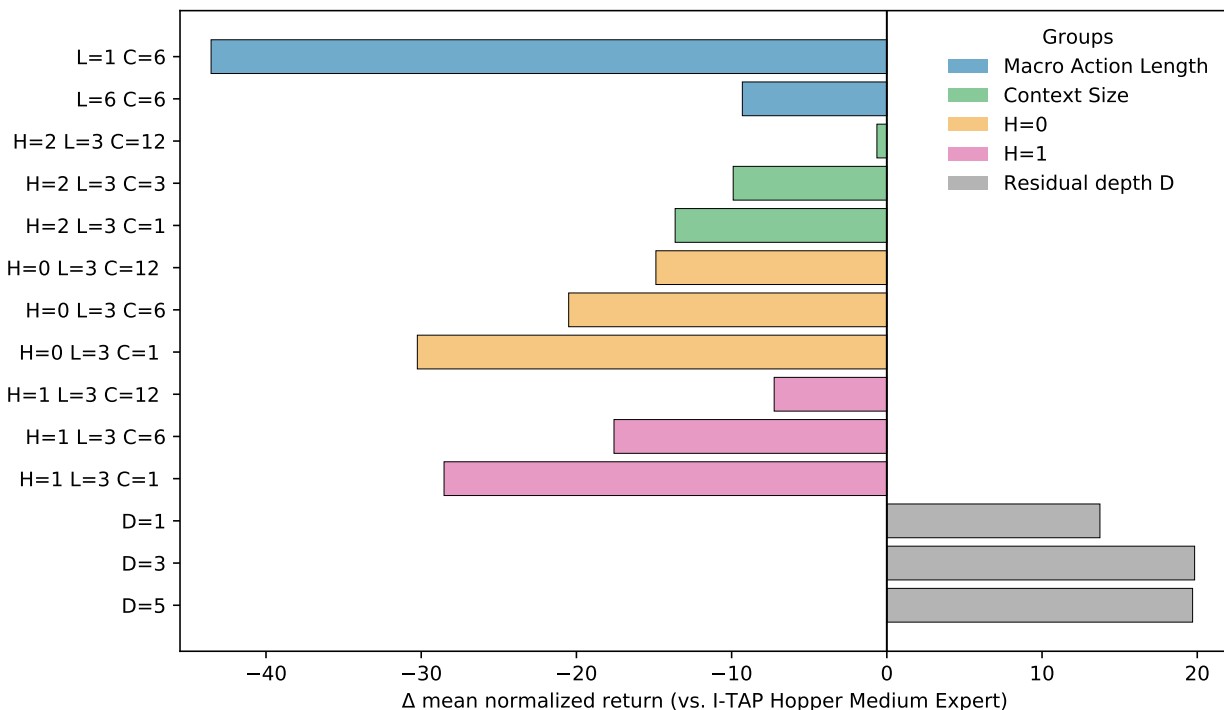

Figure 4: Ablation results across Adroit (expert) and MuJoCo Hopper. We plot $\Delta$ scores relative to I-TAP on Hopper medium–expert, which serves as the baseline (zero line).

**Adroit Control** Adroit poses high-dimensional state–action spaces and fine-grained control demands, and provides partial-observability evaluation across diverse manipulation tasks: Pen, Hammer, Door, and Relocate. Table 3 shows that I-TAP remains strongest on average in both the fully observable and partially observable expert settings. On cloned datasets, I-TAP obtains the best results on Pen and Door, while FQL is strongest on Hammer and L-MAP on Relocate. On expert datasets, I-TAP leads Hammer, Door, and Relocate, while FQL obtains the highest Pen score. Under partial observability (POMDP), I-TAP achieves the highest mean score and leads Pen, Hammer, and Relocate, while LSDT leads Door. These results highlight the benefit of context-conditioned planning when state aliasing induces apparent stochasticity, while also showing that sequence and model-free baselines can be competitive on individual Adroit tasks. We use residual depth D=2 for cloned and D=3 for expert datasets; the stronger expert results are consistent with the intuition that coarse-to-fine residual quantization helps retain the granularity needed for high-dimensional continuous control.

## 5.2 Ablation Study

We present analyses and ablations of macro action length, context length, planning horizon, and residual depth. Figure 4 summarizes the results from ablation studies conducted across deterministic, moderate-noise, and high-noise Mujoco Hopper control tasks and Adroit (expert) tasks.

**Macro Action Length.** We vary the macro length $L$ and compare $L=1$ versus $L=6$. Increasing to $L=6$ moderately degrades performance, whereas decreasing to $L=1$ causes a substantially larger drop. Two factors explain this: (i) with fixed $C$, the *effective history* observed by the planner scales as $C \times L$ steps, so $L=1$ shortens the usable history and weakens in-context adaptation; and (ii) shorter macros increase the *branching factor* during search and make the RQ-VAE more prone to *overfitting*, which requires careful hyperparameter control.

**Context Size.** We ablate context size $C \in \{1, 3, 12\}$ (in latent tokens). The datasets are produced under distinct latent dynamics; thus, context is useful for *inferring the active mode* and for feeding back recent

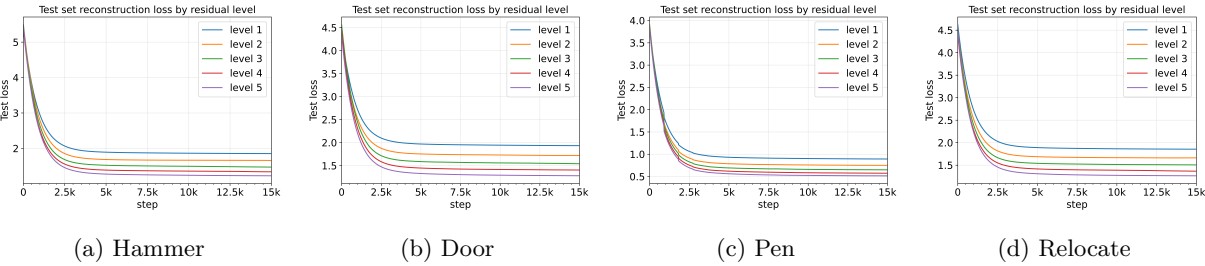

(a) Hammer      (b) Door      (c) Pen      (d) Relocate

Figure 5: Test reconstruction losses across residual levels.

rewards to steer exploration. Increasing context from $C=1$ (approximately 3 past transitions when $L=3$) to $C=12$ (approximately 36 transitions) yields consistent gains. This demonstrates that short, latent contexts already help (few tokens cover many steps), and performance improves as $C$ grows because the model better disambiguates latent parameters and aggregates noisy feedback.

**Planning Horizon.** We vary the MCTS look-ahead depth $H$ (in latent tokens). With macro length $L=3$, a depth $H$ corresponds to $H \times L$ planning horizon in the raw action space. Removing planning ($H=0$) harms performance for all $C$; the drop is smaller on deterministic tasks where the in-context prior is already strong, but it remains insufficient under stochasticity. Increasing to $H=1$ yields consistent gains across settings; at $H=2$, we observe a pronounced jump, a shorter-context model ($C=6$) matches a longer-context baseline ($C=12$). This supports our hypothesis that MCTS can *mitigate uncertainty* due to partial observability and stochastic dynamics by taking expectations over futures: deeper look-ahead can substitute for additional context up to a point before compute and diminishing returns dominate.

**Residual Depth** We vary the residual quantization depth $D$ for RQ-VAE. Reconstruction error *decreases monotonically* with $D$ (Figure 5), with diminishing returns: the largest drop is from $D=1$ to $D=3$. Control performance mirrors this: the task-averaged score rises from 113.40 at $D=1$ to 119.50 at $D=3$, and then plateaus (119.37 at $D=5$). This highlights that increasing residual depth improves reconstruction quality for high-dimensional features and shows that $D=3$ preserves action granularity for high-DoF control without incurring the computation/optimization overhead of very deep stacks.

## 5.3   Decision Time vs Context Length

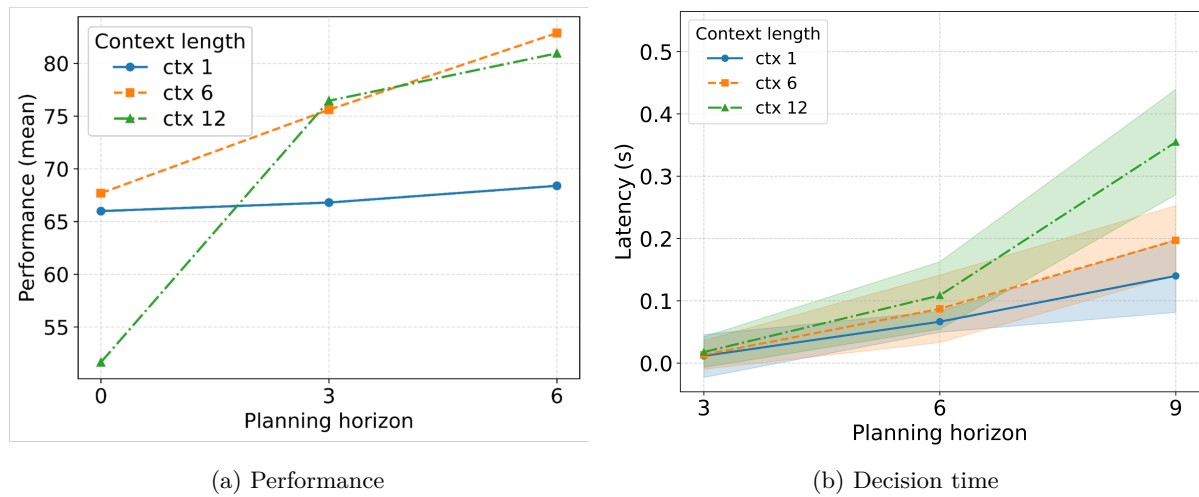

(a) Performance      (b) Decision time

Figure 6: Performance and decision-time trade-off across planning horizon and context length. The left plot reports normalized performance, while the right plot reports average per-decision latency with 32 action candidates.

Figure 6 shows the trade-off between context size, planning depth, performance, and decision-making efficiency. Increasing the planning horizon generally improves performance, while longer contexts and deeper planning increase the average decision time.

Thus, in scenarios characterized by dynamics influenced by unobserved states, our findings underline the necessity of carefully selecting context lengths. It is crucial to encode historical observations effectively, extracting the most informative signals without redundant details that contribute minimal additional predictive power but significantly raise computational latency. In practical applications, particularly those involving stochasticity and continuous observation-action spaces, balancing sufficient context length to maintain accuracy against the latency imposed by deeper planning becomes a critical design decision. Efficient decision-making, therefore, relies on identifying a context size that captures the essential dynamics without incurring unnecessary computational overhead.

## 6 Related Works

**Reinforcement Learning as Sequence Modeling.** Recent advances have reframed reinforcement learning (RL) as a sequence modeling problem, initiated by Decision Transformer (DT), which formulates RL as supervised learning conditioned on desired returns (Chen et al., 2021). Building on these ideas, Algorithm Distillation (Laskin et al., 2023) and Decision-Pretrained Transformer (Lee et al., 2023) leverage Transformers to distill optimal behaviors from historical trajectories, enabling rapid in-context adaptation. More recently, hierarchical variants like In-context Decision Transformer (IDT; Huang et al. (2024)) extend this paradigm by modeling high-level decisions, alleviating computational bottlenecks associated with long context windows. However, as noted by Son et al. (2025), these methods may replicate suboptimal behaviors due to the absence of explicit planning mechanisms, a limitation potentially addressed by model-based planning. Moreover, supervised RL methods typically assume deterministic or near-deterministic datasets, inherently limiting their effectiveness in stochastic environments, where conditioning solely on outcomes can lead to incorrect decisions (Paster et al., 2022). Unlike prior works relying on large contexts in near-deterministic settings, our approach explicitly targets efficient adaptation in dynamic, stochastic environments using a limited context window.

**Model-Based Reinforcement Learning.** From a model-based perspective (Antonoglou et al., 2022; Schrittwieser et al., 2020), Janner et al. (2021) introduced beam search over a Transformer dynamics model for planning, inspiring subsequent methods that plan in learned latent spaces. In particular, TAP (Jiang et al., 2023) and L-MAP (Luo et al., 2025) employ temporal abstraction by encoding multi-step action segments into discrete codes via state-conditioned VQ-VAEs, then planning over these compact tokens. While beam search with a learned model is effective in largely deterministic settings, L-MAP further adopts MCTS to handle stochastic dynamics and improve robustness. Nevertheless, both methods assume full observability, which can limit performance under state aliasing. Our proposed I-TAP bridges this gap by conditioning planning on recent histories to mitigate partial observability and use MCTS to take expectations over possible futures and deviate from suboptimal priors.

## 7 Discussion

We presented the *In-Context Latent Temporal Abstraction Planner* (I-TAP), an offline RL approach that discretizes observation–macro-action trajectories and performs planning in a learned latent space. By combining temporal abstraction with in-context conditioning, I-TAP reduces planning complexity while enabling adaptation to latent regime variations through the test-time context window. Residual quantization provides a coarse-to-fine code stack to extend the scalability of the proposed method, and our factorized prior keeps planning-time inference efficient when the model is queried. Across stochastic MuJoCo and high-dimensional Adroit domains, I-TAP achieves competitive performance relative to strong offline baselines. More broadly, the same idea of discretized latent representations plus planning-compatible sequence priors may be useful beyond offline RL, for example in imitation learning settings that require temporally extended actions while retaining fine-grained control.

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

Table 4: Conceptual comparison between TAP, L-MAP, and I-TAP. The key distinction is that I-TAP changes the planner from a current-state-conditioned latent planner to a context-conditioned latent planner for stochastic and partially observable offline RL.

| Axis | TAP | L-MAP | I-TAP (ours) |
|---|---|---|---|
| **Problem formulation** | Offline RL with deterministic MDP | Offline RL with stochastic MDP | Offline RL / offline meta-RL under POMDPs or unobserved latent regimes |
| **Planner decision node** | Current state | Current state | Current observation plus recent latent interaction history |
| **Latent action representation** | Single-code latent option | Single-code latent macro-action | Residual code stack for each macro-action token |
| **Quantization** | State-conditioned VQ-VAE | State-conditioned VQ-VAE | Observation/context-conditioned RQ-VAE with depth $D$ |
| **Prior / dynamics conditioning** | State-conditioned latent trajectory model | $p(z \mid s)$ and $p(z' \mid z, s)$ | $p(k_{t,1:D} \mid k_{<t,1:D}, o_t)$, with temporal and depth factorization |
| **Search algorithm** | Beam / latent trajectory search | MCTS with UCT and progressive widening | MCTS with context-conditioned P-UCT over residual code stacks |
| **Role of learned prior** | Proposes likely latent action trajectories | Samples plausible macro-actions and latent futures from the current state | Guides search using a context-conditioned prior while allowing MCTS to override suboptimal priors |
| **Return handling** | Return/reward information used in trajectory scoring | RTG masked for latent code assignment to reduce return-induced fragmentation | Long-horizon RTG masked; short-horizon return retained in context for latent-regime inference |
| **Test-time adaptation** | No explicit history-conditioned adaptation | No explicit history-conditioned adaptation | In-context adaptation from recent history, without gradient updates |
| **Partial observability** | Not explicitly modeled | Not explicitly modeled | Explicitly modeled through finite-history context as a belief proxy |
| **Held-out latent regimes** | Not explicitly targeted | Robustness/generalization through MCTS, but no explicit in-context adaptation mechanism | Explicitly targeted through context-conditioned prior and planning |

## A   Appendix

**Difference from TAP and L-MAP.**   TAP and L-MAP demonstrate that learned temporal abstractions can make offline planning tractable in high-dimensional continuous control. However, both methods plan from a current-state-conditioned latent representation and do not explicitly use recent interaction history as part of the planner state. I-TAP instead treats recent history as a finite-context belief proxy and performs MCTS over context-conditioned residual code stacks. This changes the role of the sequence model: it is not only a prior over likely latent actions, but also a context-conditioned latent dynamics model used inside search. Residual quantization is therefore not introduced as a standalone novelty; it provides the capacity needed to represent high-dimensional, context-dependent observation–macro-action segments while retaining a compact discrete interface for planning.

Table 5: Self-contained summary of the main I-TAP components and their roles in the planning pipeline.

| Component | Intuition | Why it is needed | How I-TAP uses it |
|---|---|---|---|
| Latent macro-action | A discrete representation of a short continuous action segment. | Planning over primitive continuous actions has large branching factor and long horizon. | Each candidate tree edge corresponds to a latent macro-action token or residual-code stack. |
| Residual quantization | Represents one macro-step by a coarse-to-fine stack of discrete codes. | A single VQ code may not preserve enough granularity for high-dimensional observation–action segments. | RQ-VAE maps each macro token to $k_{b,1:D}$, increasing effective capacity while keeping planning discrete. |
| Context window | A short recent history used as a finite-history belief proxy. | The current observation may not reveal the latent regime or hidden state. | The Transformer conditions on recent residual-code history and the current observation. |
| Return masking | Prevents latent codes from being determined by noisy long-horizon outcomes. | In stochastic offline data, high RTG can reflect luck rather than reliable action quality. | I-TAP masks long-horizon RTG and keeps short-horizon return only in context as a regime signal. |
| Temporal prior | A sequence model over residual-code stacks. | MCTS needs likely candidate actions and possible future outcomes. | The Transformer provides both a context-conditioned action prior and a latent transition model. |
| MCTS over latent tokens | Searches over discrete macro-action tokens instead of raw continuous actions. | Direct continuous-action planning is expensive under stochasticity. | MCTS samples multiple latent futures, backs up expected values, and can override suboptimal priors. |

## A.1 Latent Search Tree Construction

**Algorithm 1** $\textsc{SampleStack}(p_\phi, \mathcal{C}, D, \boldsymbol{\xi}, \boldsymbol{\rho})$

1: $\mathbf{u}_1 \leftarrow$ scores from $p_\phi(k_1 \mid \mathcal{C})$
2: $k_1 \sim \textsc{TopKTempCat}(\mathbf{u}_1, \xi_1, \rho_1)$
3: $K \leftarrow (k_1)$
4: **for** $d = 2$ **to** $D$ **do**
5: $\quad \mathbf{u}_d \leftarrow$ scores from $p_\phi(k_d \mid K_{1:d-1}, \mathcal{C})$
6: $\quad k_d \sim \textsc{TopKTempCat}(\mathbf{u}_d, \xi_d, \rho_d)$
7: $\quad K \leftarrow (K, k_d)$                                      ▷ append
8: **end for**
9: **return** $K$

**Algorithm 2** Pre-constructing the Latent Search Space (Residual Stack, Macro Tokens)

**Require:** Current observation $o_{t_k}$; context $\big((G_{t-cL}^{(L)}, o_{t-cL}, a_{t-cL}), \ldots, (G_{t-1}^{(L)}, o_{t-1}, a_{t-1})\big)$; encoder $f_{\text{enc}}$; decoder $f_{\text{dec}}$; residual-stack model $p_\phi$; residual depth $D$; per-depth temperatures $\boldsymbol{\xi} = (\xi_1, \ldots, \xi_D)$; per-depth top truncation $\boldsymbol{\rho} = (\rho_1, \ldots, \rho_D)$; # coarse samples $M$; # residual completions per coarse sample $J$; # lookahead samples $N$; tree depth $H$; # kept per node $\kappa_{\text{keep}}$; # proposals per node $B$

**Ensure:** Latent search tree $\mathcal{T}$ with cached promising residual-stack codes

1: **Encode macro-context**
2: $k_{t_{k-1}:t_{k-c}, 1:D} \leftarrow f_{\text{enc}}\big((G_{t-cL}^{(L)}, o_{t-cL}, a_{t-cL}), \ldots, (G_{t-1}^{(L)}, o_{t-1}, a_{t-1})\big)$
3: Initialize tree $\mathcal{T}$ with root node $s_{t_k} = (o_{t_k}, k_{t_{k-1}:t_{k-c}, 1:D})$
4: $\mathcal{C}_k \leftarrow (o_{t_k}, k_{t_{k-1}:t_{k-c}, 1:D})$
5: **Step 1: sample and score initial macro stacks at index $k$**
6: /* $M$ coarse draws for depth 1; for each, $J$ residual completions to depth $D$ */
7: $\mathbf{u}_{k,1} \leftarrow$ scores from $p_\phi\big(k_{t_k,1} \mid \mathcal{C}_k\big)$
8: **for** $i = 1$ **to** $M$ **(parallel) do**
9: $\quad k_{t_k,1}^{(i)} \sim \textsc{TopKTempCat}\big(\mathbf{u}_{k,1}, \xi_1, \rho_1\big)$
10: $\quad$ **for** $j = 1$ **to** $J$ **(parallel) do**
11: $\quad\quad K_{t_k}^{(i,j)} \leftarrow (k_{t_k,1}^{(i)})$
12: $\quad\quad$ **for** $d = 2$ **to** $D$ **do**
13: $\quad\quad\quad \mathbf{u}_{k,d} \leftarrow$ scores from $p_\phi\big(k_{t_k,d} \mid K_{t_k,1:d-1}^{(i,j)}, \mathcal{C}_k\big)$
14: $\quad\quad\quad k_{t_k,d}^{(i,j)} \sim \textsc{TopKTempCat}\big(\mathbf{u}_{k,d}, \xi_d, \rho_d\big)$
15: $\quad\quad\quad K_{t_k}^{(i,j)} \leftarrow (K_{t_k}^{(i,j)}, k_{t_k,d}^{(i,j)})$                  ▷ append
16: $\quad\quad$ **end for**
17: $\quad\quad z_{t_k}^{(i,j)} \leftarrow \text{Embed}\big(K_{t_k}^{(i,j)}\big)$
18: $\quad\quad \widehat{G}_{t_k}^{(L)}\big(K_{t_k}^{(i,j)}\big) \leftarrow$ current-step head from $f_{\text{dec}}$
19: $\quad\quad$ **for** $n = 1$ **to** $N$ **(parallel) do**
20: $\quad\quad\quad \mathcal{C}' \leftarrow (o_{t_k}, K_{t_k}^{(i,j)}, k_{t_{k-1}:t_{k-c}, 1:D})$
21: $\quad\quad\quad K_{t_{k+1}}^{(i,j,n)} \leftarrow \text{SampleStack}(p_\phi, \mathcal{C}', D, \boldsymbol{\xi}, \boldsymbol{\rho})$
22: $\quad\quad\quad z_{t_{k+1}}^{(i,j,n)} \leftarrow \text{Embed}\big(K_{t_{k+1}}^{(i,j,n)}\big)$
23: $\quad\quad\quad \hat{y}_{t_{k+1}}^{(i,j,n)} \leftarrow f_{\text{dec}}\big(z_{t_k}^{(i,j)}, z_{t_{k+1}}^{(i,j,n)}, o_{t_k}, k_{t_{k-1}:t_{k-c}, 1:D}\big)$
24: $\quad\quad$ **end for**
25: $\quad\quad \text{score}\big(K_{t_k}^{(i,j)}\big) \leftarrow \frac{1}{N} \sum_{n=1}^{N} \Big(\widehat{G}_{t_k}^{(L)}\big(K_{t_k}^{(i,j)}\big) + \big[\hat{y}_{t_{k+1}}^{(i,j,n)}\big]_{\text{rtg}}\Big)$
26: $\quad\quad \bar{y}_{t_{k+1}}^{(i,j)} \leftarrow \frac{1}{N} \sum_{n=1}^{N} \hat{y}_{t_{k+1}}^{(i,j,n)}; \quad \hat{o}_{t_{k+1}}^{(i,j)} \leftarrow \text{Obs}\big(\bar{y}_{t_{k+1}}^{(i,j)}\big)$
27: $\quad$ **end for**
28: **end for**
29: Select top-$\kappa_{\text{keep}}$ stacks $\{K_{t_k}^{(i,j)}\}$ by score; for each, attach child node $\big(\hat{o}_{t_{k+1}}^{(i,j)}, K_{t_k}^{(i,j)}\big)$ under the root in $\mathcal{T}$
30: **Step 2: recursive latent-tree expansion over macro indices**
31: **for** $h = 2$ **to** $H$ **do**
32: $\quad$ Let $\mathcal{N}_{h-1}$ be the nodes at depth $h-1$ of $\mathcal{T}$
33: $\quad$ **for each** node $(\hat{o}, K^c) \in \mathcal{N}_{h-1}$ **(parallel) do**
34: $\quad\quad \mathcal{C} \leftarrow (\hat{o}, K^c, k_{t_{k-1}:t_{k-c}, 1:D})$
35: $\quad\quad$ **for** $b = 1$ **to** $B$ **(parallel) do**
36: $\quad\quad\quad K_{t_{k+h-1}}^{(b)} \leftarrow \text{SampleStack}(p_\phi, \mathcal{C}, D, \boldsymbol{\xi}, \boldsymbol{\rho})$
37: $\quad\quad\quad z_{t_{k+h-1}}^{(b)} \leftarrow \text{Embed}\big(K_{t_{k+h-1}}^{(b)}\big)$

38:  $\widehat{G}^{(L)}_{t_{k+h-1}}\big(K^{(b)}_{t_{k+h-1}}\big) \leftarrow$ current-step head from $f_{\text{dec}}$
39:  **for** $n = 1$ **to** $N$ **(parallel) do**
40:  $\quad \mathcal{C}^\star \leftarrow (\hat{o},\, K^{(b)}_{t_{k+h-1}},\, k_{t_{k-1}:t_{k-c},\,1:D})$
41:  $\quad K^{(b,n)}_{t_{k+h}} \leftarrow \text{SampleStack}(p_\phi,\, \mathcal{C}^\star,\, D,\, \boldsymbol{\xi},\, \boldsymbol{\rho})$
42:  $\quad z^{(b,n)}_{t_{k+h}} \leftarrow \text{Embed}\big(K^{(b,n)}_{t_{k+h}}\big)$
43:  $\quad \hat{y}^{(b,n)}_{t_{k+h}} \leftarrow f_{\text{dec}}\big(z^{(b)}_{t_{k+h-1}},\, z^{(b,n)}_{t_{k+h}},\, \hat{o},\, k_{t_{k-1}:t_{k-c},\,1:D}\big)$
44:  **end for**
45:  $\text{score}\big(K^{(b)}_{t_{k+h-1}}\big) \leftarrow \frac{1}{N} \sum_{n=1}^{N} \Big(\widehat{G}^{(L)}_{t_{k+h-1}}\big(K^{(b)}_{t_{k+h-1}}\big) + \big[\hat{y}^{(b,n)}_{t_{k+h}}\big]_{\text{rtg}}\Big)$
46:  $\bar{y}^{(b)}_{t_{k+h}} \leftarrow \frac{1}{N} \sum_{n=1}^{N} \hat{y}^{(b,n)}_{t_{k+h}};\quad \hat{o}^{(b)}_{t_{k+h}} \leftarrow \text{Obs}\big(\bar{y}^{(b)}_{t_{k+h}}\big)$
47:  **end for**
48:  Select top-$\kappa_{\text{keep}}$ from $\{K^{(b)}_{t_{k+h-1}}\}^B_{b=1}$ by score and attach as children $(\hat{o}^{(b)}_{t_{k+h}},\, K^{(b)}_{t_{k+h-1}})$ under $(\hat{o}, K^c)$ in $\mathcal{T}$
49:  **end for**
50: **end for**
51: **return** $\mathcal{T}$

## A.2 Experiment Details

### A.2.1 Implementation Details

The hyperparameter settings for I-TAP are listed in Table 6. For baselines, we follow the implementations and recommended hyperparameters from the original works, including TAP (Jiang et al., 2023), L-MAP (Luo et al., 2025), CQL (Kumar et al., 2020), IQL (Kostrikov et al., 2022), DT (Chen et al., 2021), LSDT (Wang et al., 2025), FQL (Park et al., 2025), and 1R2R (Rigter et al., 2023). For the Stochastic MuJoCo benchmarks, the results of TAP, 1R2R, CQL, and IQL are taken from the L-MAP paper (Luo et al., 2025), which evaluates these methods under the same stochastic-domain protocol. We train and evaluate LSDT and FQL under the same pooled stochastic-domain protocol as I-TAP, DT, and L-MAP, and use the FQL paper's reported standard Adroit results where applicable. For deterministic domains, the CQL and IQL results are taken from their respective papers (Kumar et al., 2020; Kostrikov et al., 2022). Each run of I-TAP takes approximately 6 hours on one NVIDIA RTX 5090 GPU with an Intel(R) Core(TM) i9-14900KS CPU.

### A.2.2 Domains

**Stochastic MuJoCo.** The Stochastic MuJoCo tasks introduced by Rigter et al. (2023) apply incremental perturbation forces following a uniform random walk (Popko et al., 2016). At each timestep, the perturbation force $f_t$ is updated as:

$$f_{t+1} = f_t + \Delta f, \quad \Delta f \sim \text{Uniform}\left(-0.1 \cdot f_{\text{MAX}},\, 0.1 \cdot f_{\text{MAX}}\right), \tag{1}$$

with the total perturbation clipped to remain within $[-f_{\text{MAX}}, f_{\text{MAX}}]$. This model introduces persistent, incremental perturbations, representing a baseline scenario. For both Hopper and Walker2D environments, the perturbation force $f_t$ is applied horizontally along the x-axis to simulate external disturbances, such as wind gusts. The maximum perturbation magnitude $f_{\text{MAX}}$ specifically for the Hopper moderate perturbation level is 2.5 Newtons, high perturbation level is 5 Newtons, Walker2D moderate perturbation level is 7 Newtons, high perturbation level is 12 Newtons.

**Adroit as a POMDP.** Many Adroit manipulation tasks expose *privileged* channels that directly encode the goal state and progress (e.g., target poses, object target deltas, insertion depth). For the POMDP Adroit experiments, the same observation mask is applied during both offline training and evaluation for all methods. Concretely, let the environment produce an observation $o_t \in \mathbb{R}^d$ under the standard `v1` layout. We introduce a fixed binary mask $m \in \{0, 1\}^d$ (zeros at privileged indices) and set

$$\tilde{o}_t = m \odot o_t, \qquad O(\tilde{o}_t \mid s_t) = \delta\big(\tilde{o}_t - m \odot o_t\big),$$

yielding a POMDP in which progress toward the goal must be *inferred* from history rather than read off directly.

**Masking regimes.** Indices refer to $0$-based positions in the default `v1` observation vector.

Table 6: List of Hyper-parameters

| Environment | Hyper-parameter | Value |
|---|---|---|
| All | learning rate | $1 \times 10^{-4}$ |
| All | batch size | 512 |
| All | dropout probability | 0.1 |
| All | number of attention heads | 4 |
| All | macro action length $L$ | 3 |
| All | embedding size (latent code) | 512 |
| All | $c_1$ | 1.25 |
| All | $c_2$ | 19652 |
| All | $\alpha_{\text{tail}}$ | 1 |
| All | $\alpha_{\text{ctx}}$ | 0.1 |
| All | $\beta_{\text{ps}}$ | 1 |
| MuJoco | context length $c$/training sequence length | 6/24 |
| MuJoco | discount factor | 0.99 |
| MuJoco | number of Transformer layers | 4 |
| MuJoco | feature vector size | 512 |
| MuJoco | codebook size | 512 |
| MuJoco | initial number of policy samples $M$ | 16 |
| MuJoco | number of transition samples $N$ | 4 |
| MuJoco | number of policy samples $B$ | 4 |
| MuJoco | number of MCTS iterations | 100 |
| Mujoco | $\kappa_{\text{keep}}$ | 50% |
| MuJoco | temperature | 2 |
| MuJoco | Residual Depth | 1 |
| MuJoco | primitive planning horizon | 9 |
| Adroit | context length $c$/training sequence length | 6/24 |
| Adroit | discount factor | 0.99 |
| Adroit | number Transformer layers | 4 |
| Adroit | feature vector size | 256 |
| Adroit | codebook size | 512 |
| Adroit | initial number of policy samples $M$ | 16 |
| Adroit | number of transition samples $N$ | 4 |
| Adroit | number of policy samples $B$ | 4 |
| Adroit | number of MCTS iterations | 100 |
| Adroit | Residual Depth | 2,3 |
| Adroit | $\kappa_{\text{keep}}$ | 10% |
| Adroit | $J$ | 4 |
| Adroit | temperature | 1 |
| Adroit | primitive planning horizon | 9 |

- **Pen** ($d$=45): $\mathcal{I} = \{36, 37, 39, 40, 42, 43\}$.

- **Relocate** ($d$=39): $\mathcal{I} = \{30, \ldots, 38\}$.

- **Door** ($d$=39): $\mathcal{I} = \{27, 28, 32, \ldots, 38\}$.

- **Hammer** ($d$=46): $\mathcal{I} = \{43, 44, 45\}$.

This construction leaves proprioception and contact signals intact while removing privileged goal vectors and progress proxies, thereby converting the original fully observable tasks into history-dependent POMDPs that better reflect realistic sensing and require temporal credit assignment and state estimation.

