# OpenReview forum: "In‑Context Planning with Latent Temporal Abstractions"
_TMLR — Under review for TMLR_

### Review · Reviewer_D4Ec · 2026-03-18

**Summary Of Contributions:**

The paper introduced a method named In-Context Latent Temporal-Abstraction Planner (I-TAP), an offline reinforcement learning technique that combines in-context adaptation with planning by operating in a learned discrete latent space of temporally abstract actions. It uses a residual-quantized VAE (RQ-VAE) to compress observation–macro-action sequences into stacks of discrete tokens and trains a Transformer to model their dynamics conditioned on recent context, enabling the agent to implicitly infer latent environment factors from history. At test time, the method performs Monte Carlo Tree Search directly over these latent tokens, using the learned model as both a prior and dynamics predictor, and then decodes selected plans into executable actions. Experimental results are presented for robotic locomotion tasks from the D4RL benchmark.

### Strengths

- The paper provides a well-motivated formulation of combining in-context adaptation with planning under partial observability.

- The method is a coherent integration of RQ-VAE, Transformer-based sequence modeling, and MCTS planning.

- Although only three random seeds were considered, the method seems to shows good empirical performance across stochastic and partially observable environments.

### Weaknesses

- The method is incremental relative to prior work (e.g., L-MAP/TAP), with improvements mainly in components rather than in the core paradigm. In particular, the proposed method differs from previous methods by conditioning on a short history (context window) in order to deal with partially-observable and stochastic problems.

- The use of RQ-VAE in place of a standard VQ-VAE appears to be an incremental contribution rather than a fundamental innovation. While residual quantization increases representational capacity through a stack of discrete codes and can improve reconstruction fidelity, it follows a well-established extension of vector quantization rather than introducing a new modeling paradigm. In the context of this work, RQ-VAE mainly serves as a stronger encoder for latent temporal abstractions, but it does not substantially change the overall framework of learning discrete actions and planning over them.

- The paper relies on several concepts that are not fully explained in a self-contained manner, requiring readers to refer to prior work—particularly L-MAP—to understand key components. For example, the use of latent macro-actions and planning in a discretized latent space is introduced without a detailed standalone explanation, even though it is central to the method (Sections 3–4) . Similarly, the MCTS procedure over latent tokens (Section 4.3) assumes familiarity with how latent action spaces are constructed and searched in prior work, with limited intuition provided for how this differs in practice. The residual quantization mechanism (Section 3) is briefly described but depends heavily on prior literature, and important design choices—such as return masking and the structure of token representations (Section 4.1)—are not thoroughly motivated. As a result, understanding the full pipeline often requires cross-referencing earlier methods like L-MAP.

- The paper does not compare to any method specifically tailored to deal with partial observability. If would be relevant to include a baseline (e.g., [1]) that employs a recurrent architecture or some other technique to tackle such problems.

[1] Long-Short Decision Transformer: Bridging Global and Local Dependencies for Generalized Decision-Making. Jincheng Wang and Penny Karanasou and Pengyuan Wei and Elia Gatti and Diego Martinez Plasencia and Dimitrios Kanoulas. {The Thirteenth International Conference on Learning Representations, 2025. https://openreview.net/forum?id=NHMuM84tRT

**Audience:**

Yes

**Audience Explanation:**

Researchers working on model-based RL, in-context learning, or planning with learned representations would find the proposed integration of transformer-based priors with latent-space MCTS potentially relevant, even if some aspects of novelty or evaluation may limit its broader appeal.

**Claims And Evidence:**

No

**Claims Explanation:**

- The paper claims to empirically outperform strong baselines. However, the experiments were repeated with only three random seeds. Hence, it is not clear whether the results are statistically significant. Moreover, in Table 1 for instance, the mean +- std intervals overlap in many cases (e.g., Medium-Expert Walker-2D), although the performance of the proposed method is written in bold. Typically, only statistically significant results are presented in bold.

- The paper’s claim of adaptation is not fully supported by strong out-of-distribution evidence. While I-TAP demonstrates improved performance across varying stochastic regimes, these regimes are largely drawn from the same training distribution or involve mild interpolation between seen settings . As a result, the experiments primarily evaluate in-distribution adaptation rather than generalization to genuinely novel tasks or dynamics. Moreover, the work does not provide explicit analysis of how the model adapts to new conditions or comparisons to alternative approaches for handling partial observability, making it difficult to attribute the gains to true in-context adaptation rather than improved modeling and planning within a familiar setting.

**Requested Changes:**

- The current experiments mainly demonstrate in-distribution adaptation. The authors should include stronger out-of-distribution evaluations (e.g., unseen environments, task variations, or more extreme regime shifts) or provide deeper analysis to support claims of adaptation.


- The paper should compare against methods explicitly designed for partial observability, such as recurrent or belief-state approaches (e.g., LSTM-based policies or Transformer variants like Long-Short Decision Transformer), to better isolate the contribution of in-context conditioning.


- Results are based on only three random seeds, and several reported improvements appear within overlapping standard deviations. The authors should increase the number of seeds and/or include statistical significance testing, and only highlight statistically significant improvements.


- Several key components (e.g., latent macro-actions, residual quantization, return masking, and MCTS over latent tokens) should be explained more clearly and intuitively without requiring prior familiarity with L-MAP or related work.

---

> ### Author Response · Authors · 2026-06-05
> **Response 1 for Reviewer D4Ec**
>
> We deeply appreciate the reviewer's insightful feedback. Please find our detailed responses to the comments below. We are available to engage in further discussions should any additional questions arise.
>
> **Response to OOD/adaptation concern.**
> We thank the reviewer for raising this point. To further emphasize the out-of-distribution aspect, we have revised the experimental discussion to explicitly separate **(i) in-distribution latent-regime adaptation**, **(ii) held-out latent-regime generalization**, and **(iii) partial-observability evaluation across diverse Adroit manipulation tasks**.
>
> - **Held-out latent-regime evaluation.**
>
> Although Table 1 evaluates regimes included in the pooled offline training data, Table 2 evaluates **latent-parameter values not included during training**. For Hopper, the held-out regimes include \(1.25\), \(3.75\), and \(7.5\), where \(7.5\) is an extrapolation beyond the training range \(\{0,2.5,5\}\). For Walker2D, the held-out regimes include \(3.5\), \(9.5\), and \(15\), where \(15\) is an extrapolation beyond the training range \(\{0,7,12\}\). Thus, Table 2 is not an in-distribution adaptation test; it evaluates whether a single offline-trained model can adapt to unseen latent dynamics at test time.
>
> - **Empirical evidence.**
>
> On held-out latent regimes, I-TAP obtains the highest mean score across all dataset types in Table 2: Medium-Expert, Medium, and Medium-Replay. These gains are consistent across interpolation and extrapolation regimes and are larger relative to DT, which has context but no explicit planner, and relative to L-MAP, which plans but does not explicitly condition search on recent interaction history. This supports our central claim that **context-conditioned planning**, rather than context-only imitation or planning-only robustness, improves adaptation to unseen latent dynamics.
>
> - **Partial-observability evaluation across Adroit tasks.**
>
> We also evaluate Adroit under a partially observable variant where target-position coordinates are masked. This setting introduces state aliasing and therefore directly tests whether recent history helps disambiguate the underlying state. I-TAP achieves the best mean performance on the POMDP Adroit tasks and outperforms both L-MAP and TAP across the partially observable Adroit domains. This provides an additional evaluation beyond stochastic MuJoCo latent-parameter shifts across diverse manipulation tasks: Pen, Hammer, Door, and Relocate.
>
> - **Mechanism-level support.**
>
> The ablations further support the adaptation mechanism. Increasing context length improves performance because the model can better infer the active latent regime from history. Removing planning (\(H=0\)) hurts performance, especially under stochasticity, showing that context alone is insufficient. Increasing MCTS depth improves performance because the planner evaluates multiple latent futures and backs up expected values. Together, these results support the intended decomposition: **context helps infer the regime, and MCTS uses that context-conditioned model to optimize decisions under uncertainty**.
>
> - **Clarified claim.**
>
> We clarify that our generalization/adaptation claim is not about transfer to entirely new environment families. Rather, it concerns **held-out perturbation regimes within the stochastic MuJoCo offline meta-RL setting**, together with a complementary **partial-observability evaluation on four Adroit manipulation domains**. In all cases, adaptation is performed through the recent context window without test-time gradient updates.

---

> > ### Author Response · Authors · 2026-06-05
> > **Response 2 for Reviewer D4Ec**
> >
> > **Concern: Add partial-observability baselines, especially recurrent/belief-state methods or LSDT.**
> >
> > Thank you for the suggestion. Besides DT, which was already included as a context-conditioned sequence-modeling baseline, we have added newer baselines, including LSDT and FQL. LSDT is particularly relevant to this concern because it uses long/short history to construct a stronger sequence-model policy under partial observability, while FQL adds a recent model-free offline RL baseline.
> >
> > On Adroit POMDP, I-TAP achieves a mean score of \(75.72\), compared with LSDT's \(46.24\), a \(+29.48\) absolute improvement. Per task, I-TAP is higher on Pen (\(85.97\) vs. \(17.84\), \(+68.13\)), essentially tied on Hammer (\(69.91\) vs. \(69.74\), \(+0.17\)), slightly lower on Door (\(96.42\) vs. \(97.20\), \(-0.78\)), and substantially higher on Relocate (\(50.59\) vs. \(0.16\), \(+50.43\)). Thus, LSDT is competitive on some individual POMDP tasks, but I-TAP is stronger on average and more robust across tasks, supporting the benefit of combining history conditioning with decision-time planning.
> >
> > **Concern: Results are based on only three random seeds, and several reported improvements appear within overlapping standard deviations.**
> >
> > We appreciate the reviewer's suggestion. We have updated the results to report five random seeds and revised the discussion to focus on aggregate trends rather than over-interpreting small row-level differences. The five-seed results preserve the main empirical pattern: I-TAP remains strongest on average across stochastic MuJoCo, held-out latent regimes, and Adroit POMDP settings.
> >
> > **Concern: Clarity of key components.**
> > We thank the reviewer for pointing this out. We agree that the original presentation was too compact. We have revised the manuscript to make the method more self-contained and intuitive before presenting the technical details.
> >
> > We added a new self-contained overview at the beginning of the Method section that explains I-TAP as a four-step pipeline:
> > (1) group primitive actions into macro-actions to shorten the planning horizon;
> > (2) encode each observation-macro-action segment into a discrete residual-code stack;
> > (3) learn a context-conditioned Transformer prior/dynamics model over these code stacks; and
> > (4) run MCTS directly over latent tokens, then decode the selected token stack into executable actions.
> >
> > We added a component-summary table explaining the role of latent macro-actions, RQ-VAE, return masking, the temporal prior, and MCTS. These changes are intended to make the paper understandable without requiring prior familiarity with prior works.

---

### Review · Reviewer_yqGC · 2026-04-13

**Summary Of Contributions:**

The paper presents a method for solving planning problems with offline RL datasets. The method, I-TAP, consists of a transformer model which uses a residual-quantized VAE to aggregate multi-step state observations and macro actions into discrete macro states for planning. The planning policy is generated via model-based MCTS over these aggregated states. The planning process is evaluated in a subset of Stochastic Mujoco tasks with varying levels of stochasticity and different underlying offline datasets.

**Additional Comments:**

Given the general move towards foundation models in the literature, I wonder if it would make sense to show that methods such as this can (or cannot) benefit from pre-training? Would it be possible to connect the action head to a VLM in the same style as a VLA?

**Audience:**

Yes

**Audience Explanation:**

Offline RL and transformer-based planning are both active research areas that seem

**Claims And Evidence:**

No

**Claims Explanation:**

Overall, I believe there are several issues with the paper as is that reduce the strength of its evidence. The paper contains a number of unclear statements and some notational vagueness, which I outline below. In addition, prior work by Luo et al. is extensively referenced and seems very similar to the work presented here. It would strongly benefit the paper to explain the advances over Luo et al. clearly. Reading the text, I believe the main difference stated is that I-TAP conditions on past policies. The advantages and differences of I-TAP over L-MAP are not clearly explained and the two models seem to perform very similarly to each other. A brief review of the L-MAP paper makes it seem like there are mostly small additional tweaks to the model and action selection process. It is unclear whether the performance benefits are mostly achieved through hyperparameter tuning.

**Background & Method**

"We introduce temporal abstractions via a fixed macro length" Prior work (Luo et al.) seems to already have introduced this. Please state clearly what the novel contributions are.

What is the variable $k$ used in Section 4.2? How does it differ from $z$.

**Experimental Section**

IQL and CQL are rather old baselines, it would be valuable to look at some newer method, e.g. those which include more flexible policy approximations like Flow Q Learning or IDQL. Furthermore, it is unclear why offline RL is not evaluated on the Adroid dataset or in the held out regime?

_Detailed comments for concrete claims_:
"Relative to L-MAP, I-TAP’s improvements highlight the benefit of context-guided action selection within search". The provided experiments do not really highlight this difference, it would be good to provide more targeted experiments to evaluate this claim.

"Increasing to L=6 does not noticeably degrade performance, whereas decreasing to L=1 causes a substantial drop." If I read the ablation results correctly, increasing L drops the performance by ca 8-10, close to the competitive baselines.

"In contrast, DT degrades more sharply under stochastic dynamics and lower-quality behavior data" This does not seem to be fully validated in the experimental data. DT's performance drops under uncertainty seem to be equal or smaller than those of the proposed method, but the base performance is lower.

Section 5.3: It would be helpful to show the performance improvement as well, otherwise the planning time increase is not easy to interpret.

**Requested Changes:**

Please address the feedback on the experimental section.

---

> ### Author Response · Authors · 2026-06-05
> **Response 1 for Reviewer yqGC**
>
> We greatly value the reviewer’s constructive feedback and have provided detailed, point-by-point responses below. We are available to discuss any further questions or concerns you may have.
>
> **Concern: Explain what is new about temporal abstraction, since Luo et al. already uses fixed macro length.**
>
> Thank you for pointing this out. Our intention was to introduce the concept as background for readers, not to claim fixed macro length as a new contribution. We have revised the wording from “introduce temporal abstraction” to “use temporal abstraction” where appropriate. The contribution is not the fixed macro length itself, but using temporal abstraction as the discrete interface for **context-conditioned residual code stacks and MCTS under stochastic or partially observable dynamics**.
>
> **Concern: What is the variable
>  used in Section 4.2? How does it differ from z**
>
> \(z\) denotes the continuous embedding produced before quantization, while vector quantization retrieves the nearest entries from the codebook; \(k\) denotes the resulting discrete token or residual code stack. Thus, \(z\) is the pre-quantization representation and \(k\) is the retrieved code-index representation used by the Transformer and planner.
>
> **Concern: Add newer baselines beyond CQL/IQL, especially FQL or IDQL.**
>
> Thank you for the suggestion. We have added **FQL** and **LSDT** to the revised experiments. FQL provides a recent model-free offline RL baseline, while LSDT provides a recent Decision Transformer-style sequence-modeling baseline. Both are trained under the same pooled stochastic-domain protocol as I-TAP, DT, and L-MAP.
>
> - **Summary of the updated results.**
> The overall trend remains unchanged. Compared with I-TAP, FQL is a model-free offline RL baseline and does not perform context-conditioned test-time planning; LSDT uses sequence modeling for action prediction, but it is still deployed as a direct policy rather than as a planner. On stochastic MuJoCo, I-TAP achieves the best aggregate performance across dataset qualities, both on the training perturbation regimes and on held-out interpolation/extrapolation regimes. On Adroit, FQL and LSDT are competitive on several individual tasks, but I-TAP remains strongest in the aggregate.
>
> We refer reviewers to the updated result tables for the full task-level comparisons.
>
> **Concern: Explain why offline RL baselines were not evaluated on Adroit and held-out regimes.**
>
> In the original submission, we focused on comparing against L-MAP as the strongest closely related planning baseline and used DT as a representative model-free trajectory-modeling method to illustrate the difficulty of stochastic settings without explicit planning. We have therefore augmented the experimental setting with additional newer baselines, including FQL and LSDT.
>
> **Concern: Support the claim that context-guided action selection within search improves over L-MAP with targeted evidence or ablation.**
>
> We agree that this claim should be tied more explicitly to targeted evidence. We emphasize the context-size ablation, where the planning horizon and macro length are fixed while only the context length changes. With \(H=2\) and \(L=3\), increasing context from \(C=1\) to \(C=12\) substantially improves the averaged Hopper medium-expert score.
>
> **Concern: Recheck ablation wording; L=6 appears to degrade performance by about 8-10 points.**
>
> Thank you for catching this. We agree that the previous wording was too strong. Our intended point was that \(L=6\) degrades performance much less severely than \(L=1\), not that it has no noticeable degradation. We revised the text to more accurately describe this.

---

> > ### Author Response · Authors · 2026-06-05
> > **Response 2 for Reviewer yqGC**
> >
> > **Concern: Recheck the claim that DT degrades more sharply; the DT drops may be comparable or smaller.**
> >
> > We agree that the original wording was too broad. On medium-expert data, DT does drop more sharply from deterministic to stochastic regimes: averaged over Hopper and Walker2D, DT drops from 83.13 in deterministic settings to 69.27 under moderate noise and 57.15 under high noise, whereas I-TAP drops from 105.81 to 100.57 and 87.71. However, on medium and medium-replay data, DT's absolute deterministic performance is already much lower, so its additional drop can be comparable or smaller. For example, on medium-replay, DT averages 45.73 in deterministic settings and 36.20 under high noise, while I-TAP averages 83.70 and 69.57. We have revised the claim to focus on the point that DT remains substantially lower in absolute performance under stochastic and lower-quality data.
> >
> > **Concern: For the latency/planning-time section, show performance improvement together with time cost.**
> >
> > Thanks for the suggestion. We have updated the figure/section to report the performance improvement together with the planning horizon.
> >
> > **Additional comments: possible pretraining or VLA/VLM-style extensions.**
> >
> > Thank you for the insightful suggestion! We agree that I-TAP could benefit from pretraining. Its residual action-tokenization mechanism provides a compact discrete interface for high-dimensional continuous control, and its downstream planning mechanism can compose and evaluate tokenized behaviors at decision time. This suggests a natural extension toward VLA/VLM-style robotic policies: pretrained multimodal or vision-language-action models could provide stronger priors over useful macro-action tokens, while I-TAP-style MCTS could leverage those priors online to perform downstream optimization. We believe this is a promising future-work direction.

---

### Author Response · Authors · 2026-06-05
**Response to novelty relative to TAP/L-MAP.**

**Response to novelty relative to TAP/L-MAP.**
We thank the reviewers for pointing out that the distinction from TAP/L-MAP was not sufficiently clear. We agree that **RQ-VAE itself is an established representation-learning technique**. The core contribution of I-TAP is instead the integration of **in-context adaptation, residual-quantized temporal abstraction, and decision-time MCTS** in an offline RL framework for stochastic and partially observable continuous control.

- **Difference from L-MAP/TAP.**
L-MAP addresses scalable offline planning by learning **state-conditioned VQ-VAE macro-actions** and using MCTS to plan over latent actions. I-TAP targets a different challenge: when the current observation is insufficient because the environment contains **unobserved latent regimes or partial observability**. Accordingly, I-TAP changes the planning state from a current-state-conditioned latent planner to a **context-conditioned latent planner**, where the MCTS decision node contains the current observation together with a recent latent interaction history. The selected edge is a residual code stack, and the future outcome distribution is predicted conditioned on both the current code and the recent context. Thus, the Transformer is not only a behavior prior; it is a **context-conditioned prior and latent dynamics model for search**.

- **Relation to model-free trajectory modeling.**
I-TAP is also different from directly deploying a sequence model as a policy. Context-conditioned sequence policies can adapt from history, but when used as direct action predictors, they lack an explicit decision-time optimizer and may inherit suboptimal behavior from the offline dataset. In stochastic environments, return-conditioning can also be misleading because high observed returns may reflect luck rather than robust decision (Paster, McIlraith, and Ba, 2022, *You Can't Count on Luck: Why Decision Transformers and RvS Fail in Stochastic Environments*). For this reason, I-TAP masks long-horizon return-to-go at all positions, retains only short-horizon return information in the context as a regime-inference signal, and relies on downstream MCTS to optimize decisions rather than requiring a manually chosen target return.

- **Role of RQ-VAE.**
In I-TAP, residual quantization is an enabling mechanism for the proposed planning interface. Each macro token must encode a high-dimensional, context-dependent observation–macro-action segment. A single-code VQ bottleneck can therefore lose action granularity or require an excessively large codebook. RQ-VAE provides a coarse-to-fine stack of discrete codes, increasing effective capacity while keeping the search space discrete and planning-compatible. To the best of our knowledge, I-TAP is the first offline RL planning framework that uses **residual-quantized temporal abstractions as the discrete interface for context-conditioned MCTS**.

We have revised the paper to explicitly state the above distinction and added an appendix comparison table separating TAP, L-MAP, and I-TAP along the axes of problem formulation, planner state, search algorithm, prior/dynamics conditioning, return handling, quantization, test-time adaptation, and partial-observability handling.